# Cardiac Organoids to Model and Heal Heart Failure and Cardiomyopathies

**DOI:** 10.3390/biomedicines9050563

**Published:** 2021-05-18

**Authors:** Magali Seguret, Eva Vermersch, Charlène Jouve, Jean-Sébastien Hulot

**Affiliations:** 1INSERM, PARCC, Université de Paris, F-75006 Paris, France; magali.seguret@inserm.fr (M.S.); eva.vermersch@inserm.fr (E.V.); charlene.jouve@inserm.fr (C.J.); 2CIC1418 and DMU CARTE, Assistance Publique Hôpitaux de Paris (AP-HP), Hôpital Européen Georges-Pompidou, F-75015 Paris, France

**Keywords:** tissue engineering, organoids, cardiomyopathies, cardiac disease models, drug evaluation

## Abstract

Cardiac tissue engineering aims at creating contractile structures that can optimally reproduce the features of human cardiac tissue. These constructs are becoming valuable tools to model some of the cardiac functions, to set preclinical platforms for drug testing, or to alternatively be used as therapies for cardiac repair approaches. Most of the recent developments in cardiac tissue engineering have been made possible by important advances regarding the efficient generation of cardiac cells from pluripotent stem cells and the use of novel biomaterials and microfabrication methods. Different combinations of cells, biomaterials, scaffolds, and geometries are however possible, which results in different types of structures with gradual complexities and abilities to mimic the native cardiac tissue. Here, we intend to cover key aspects of tissue engineering applied to cardiology and the consequent development of cardiac organoids. This review presents various facets of the construction of human cardiac 3D constructs, from the choice of the components to their patterning, the final geometry of generated tissues, and the subsequent readouts and applications to model and treat cardiac diseases.

## 1. Introduction

Heart failure and cardiac diseases remain leading causes of morbidity and mortality worldwide. Among the different etiologies, cardiomyopathies are a prominent cause of heart failure for young people [1] and affect millions of people in the world [2]. Most cardiomyopathies have been associated with mutations in genes coding for key elements of cardiac myocytes, including sarcomeric, cell signaling, or cell–cell contact proteins [3,4]. However, the genotype–phenotype relationships are incompletely understood and the underlying mechanisms leading to cardiomyopathies and complications are only progressively reported [5,6,7,8]. In addition, the recent emergence of small molecules targeting major contractile proteins of cardiomyocytes or genome editing techniques could quickly create opportunities to cure genetic cardiomyopathies [9,10,11,12]. There is thus a need for models that can recapitulate key features observed in human cardiomyopathies and can be used for disease modeling and therapeutic testing. Animal models are the most common models for cardiomyopathies [13] but have inherent limitations to recapitulate complex human physiopathology as well as inter-patient genetic variability [14,15]. For these reasons, the translation of targeted therapies for cardiomyopathies from animal trials to human subjects has been limited so far.

The recent development of human-induced pluripotent stem cells (hiPSCs) represents a promising step for rare disease modeling as it enables us to study patient-specific derived cardiomyocytes that carry the genotype of the patients [16,17]. Although initially explored as isolated cellular models, the cardiomyocytes derived from iPS cells have progressively been integrated to build three-dimensional (3D) constructs that enhance their maturation and better reproduce the features of human cardiac tissues [18,19]. These 3D multicellular structures are called cardiac organoids and recapitulate at least one organ function. Different types of organoids have been developed with diverse compositions and architectures in order to reproduce the organization of the native myocardium. Furthermore, these different geometries can lead to various readouts and applications: 3D-engineered cardiac tissues can be used to study the mechanisms involved in diseases and to test therapies. In addition, future developments could also lead to using those constructs as supports for regenerative medicine by allowing better graft viability and cell retention compared to cell therapy alone [20].

Here, we intended to cover key aspects of tissue engineering applied to the cardiac diseases and the consequent development of cardiac organoids. This review presents various facets of the construction of human cardiac 3D constructs from the choice of components to their patterning, the final geometry of generated tissues, and the subsequent readouts and applications to model and treat cardiomyopathies.

## 2. Existing Models of Cardiomyopathies

Cardiomyopathies are defined as a heterogeneous group of diseases of the myocardium, leading to abnormal structure and function [21]. They are usually inherited and are significant causes of heart failure and sudden cardiac death in young people [22]. Cardiomyopathies are typically classified into four main types based on their clinical phenotypes:**Hypertrophic cardiomyopathy (HCM)** is the most common form of genetic heart disease [23] and is characterized by a thickening of the left ventricle and diastolic dysfunction, myocyte hypertrophy, and disarray and by increased myocardial fibrosis [24,25].**Dilated cardiomyopathy (DCM)** is characterized by left ventricular enlargement or dilation in most patients and by systolic dysfunction [26,27].**Restrictive cardiomyopathy (RCM)** is rarer and is defined by nondilated left or right ventricles with diastolic dysfunction but near to normal systolic function. Arrhythmias and conduction abnormalities often occur [28].**Arrhythmogenic right ventricular cardiomyopathy (ARVC)** involves ventricular tachycardia or fibrillation and sudden cardiac death. It is characterized by a progressive loss of right ventricular myocardium, its replacement by fibrofatty tissue, and arrhythmias [29].

Most of these cardiomyopathies have a strong genetic component [3,4,11], which is however incompletely understood. Modeling of these cardiomyopathies is also challenging because of variability in the clinical expression, incomplete penetrance of diseases in carriers of pathogenic variants, and phenotypic overlap between different cardiomyopathies. The understanding of the underlying pathophysiological mechanisms is incomplete [30], thus urging for better models [15]. In addition, the pharmacological research for more common causes of heart failure is supported by the description of mechanisms leading to cardiomyopathies, as exemplified by the recent development of actin-myosin regulators.

Small animals (i.e., rodents) are the most common models for research on cardiomyopathies [13]. They offer the advantage of accessible genetic manipulation and in vivo modeling and provide a systemic model of diseases. However, the modeling of some cardiomyopathies as well the reproduction of the genetic variability observed in humans are difficult to achieve [25,26]. Moreover, rodents present physiological discrepancies with humans such as a higher beat rate, shorter action potential duration, and differences in regulators of contractile function [14].

Another approach to modeling cardiomyopathies is to use cellular or tissular models. The first cellular models developed were primary cells and immortalized cell lines. Primary cells are directly isolated from the tissue of interest and hence faithfully reproduce the disease biology. However, cardiovascular primary cells are difficult to maintain in culture and have a very limited lifetime in vitro. Furthermore, cardiac primary cells are difficult to obtain as cardiac biopsies are rarely available. Immortalized cells are modified cells for which the cell cycle was altered and that maintain active proliferation. Nevertheless, these cells can develop behaviors that are different from those demonstrated in vivo, thus limiting their use in cardiac modeling [31].

In this context, human pluripotent stem cells present several advantages over other cellular models, notably as they can differentiate into cardiomyocytes and represent an unlimited source of cardiac cells. Human embryonic stem cells (hESCs) were initially used to generate cardiomyocytes for biomedical research and medical applications [32], but they raised an important ethical issue concerning their sources from human embryos [33]. The recent development of human-induced pluripotent stem cells (hiPSC) circumvents this issue as these cells can be created from a variety of mature, somatic cells and can be differentiated into any cell of the adult body. These cells can be patient-specific, carrying and expressing the genetic identity of the primary donor. Therefore, the use of cardiomyocytes derived from patient-specific hiPSCs (hiPSC-CMs) is now a strategy of choice to study the mechanisms of inherited cardiomyopathies in vitro [34,35,36,37,38], to test therapies, or to develop cardiotoxicity assays [39,40,41,42,43]. However, besides their lack of maturity, a drawback of stem cell-derived cellular models is the fact that they only represent one component of the native myocardium, which has a much more complex architecture and is composed of several cell types.

To overcome these issues and to study models that are more similar to native cardiac tissue, researchers have started using hiPSCs and hESCs in association with other cell types or matrices to build 3D constructs [44,45,46,47]. These constructs aim at improving the maturation of cardiomyocytes and, above all, at reproducing the structure of the myocardium and its function, while reducing the biological complexity of the system to control changeable factors. Among those factors, we see that the cell composition and tissue organization can be adjusted. These engineered cardiac tissues (ECTs) can finally form different geometries, providing various readouts, and being suitable for different applications.

## 3. Cardiac Organoids to Reproduce the Features of Human Cardiac Tissue

### 3.1. The Physiological Features of Human Cardiac Tissue

In native heart tissue, cardiomyocytes are mechanically and electrically coupled to each other and are responsible for the contractile properties of the heart. While they represent 70 to 85% of the volume of the myocardium, recent studies indicate that they represent only 25 to 35% of the total number of cardiac cells [48]. Different subtypes of cardiomyocytes can be distinguished: sinoatrial nodal cells, atrioventricular nodal cells, atrial cells, ventricular cells, and Purkinje cells [49].Those cells act in concert with a vascular system and multiple non-myocyte cells. The non-myocyte cells in the myocardium, which thus represent 65 to 75% of the cells, comprise the following:**Endothelial cells**, which border the myocardium capillaries [48] and are essential for cardiomyocyte survival and reorganization [50];**Vascular smooth muscle cells and pericytes**, which sustain the vascular network and control the vascular tone [51];**Fibroblasts**, which play a prominent role in the constitution of the supporting extracellular matrix and contribute to cardiomyocyte electrical coupling, conduction system insulation, vascular maintenance, and stress-sensing [52,53];**Neurons**, which convey autonomic control of cardiac function through a brain–heart axis [54];**Immune cells**, such as myeloid or lymphoid cells [55,56]; andOther rarer cells that can have key roles in the regulation of specific functions such as regeneration or fibrotic cardiac remodeling [57].

The distribution of the main cell types has long been discussed but, recently, Litvivnukova et al., carried out a comprehensive study on transcriptomic data of six cardiac regions, unveiling the cellular landscape of the adult heart [58]. The distributions of the various cell types significantly differed between atrial and ventricular tissues. Moreover, a large heterogeneity in gene expressions within each cell population was observed, which underlines the complexity in the cell composition of the myocardium.

Lastly, cardiac cells are embedded in an elaborate three-dimensional network of acellular components called the extracellular matrix (ECM). Its basic structural unit providing its mechanical properties is composed of interstitial collagens, laminin, fibronectin, fibrillin, and elastin. ECM also comprises proteoglycans, glycoproteins, cytokines, growth factors, and proteases [59,60]. The ECM is a complex environment that provides alignment cues, biochemical signals, and mechanical support to myocytes. It also distributes mechanical forces throughout the myocardium [61].

This complex organization allows the human myocardium to reach twitch forces of 44 mN/mm${}^{2}$ for a strip [62]. Moreover, human native cardiac tissue reacts to mechanical and electrical cues. When stretched, it develops a higher twitch amplitude, which is referred to as the Frank Starling mechanism [63]. When paced at increasing frequencies, its twitch amplitude also increases: this is the Bowditch phenomenon or the positive force–frequency relationship (FFR) [64]. Finally, human myocardium also responds to the extracellular calcium concentration as an increase in extracellular calcium concentration induces an increase in twitch force developed by the myocardium [65,66].

### 3.2. Modeling the Cell Composition in Engineered Cardiac Tissues (ECTs)

The goal of tissue engineering is to recapitulate native tissue organization in vitro and to allow the study of cell–cell interactions and heart muscle function under normal and pathological conditions [67]. Hence, cell composition is a crucial parameter in 3D engineered heart tissues.

#### 3.2.1. Different Types of Cardiomyocytes

First, in 3D-engineered cardiac tissues, cardiomyocytes can be adult cells isolated from animal or human hearts, but as explained earlier, species differences are an issue in disease modeling and biopsies from human hearts are not easily accessible. Hence, we focus here on cardiomyocytes derived from hESCs and hiPSCs, which are now predominantly used for tissue engineering. Several protocols based on the successive activation and deactivation of pathways can now lead to cardiomyocytes with a high yield [68,69,70,71]. These cardiomyocytes are equivalent to fetal or neo-natal cardiomyocytes but can be further matured in 3D constructs or by different conditioning such as electrical or mechanical stimulation, or co-culture with stromal cells. Several subtypes of cardiomyocytes can be obtained by differentiation of hiPSCs; however, most directed differentiation protocols are enriched for a certain subtype of cardiomyocytes rather than yielding homogeneous cell populations [49]. Nevertheless, in [72], Zhao et al. obtained heteropolar cardiac tissues containing distinct atrial and ventricular ends, which allowed them to assess chamber-specific drug responses. A next step would be to produce subtype-specific cardiomyocyte populations (atrial, ventricular, nodal, and Purkinje cells) and to organize them spatially on a 3D construct to obtain organoids presenting the same ratios as in native tissues.

#### 3.2.2. Different Types of Non-Myocyte Cells

In order to obtain a cellular complexity similar to native tissues and to study the interactions between different cell types, non-myocytes are often added in engineered cardiac tissues. According to the chosen geometry and scaffold, the ratios of the different cell types have to be optimized to obtain a compact and functional tissue [73,74]. Cardiac fibroblasts have been shown to improve cardiomyocyte maturation and viability and to modulate the organization and synchronous beating of 3D engineered tissues [75,76,77,78]. The addition of endothelial cells has been proven to promote the formation of blood-capillary-like networks in 3D tissues [73,78]. These structures are crucial in case of in vivo transplantation as they can integrate with the host coronary circulation [79] and can therefore ensure the viability of the graft. Endothelial cells added in 3D engineered tissues also secrete autocrine and paracrine molecules, which improve cardiomyocyte survival, proliferation, and maturation [80,81,82]. Finally, Giacomelli et al. showed the importance of the addition of stromal cells in 3D cardiac constructs used in vitro for disease modeling and drug testing [76]. Indeed, with 3D cardiac organoids built from healthy hiPSC-CMs, healthy hiPSC-ECs, and mutated hiPSC-CFs, they could recapitulate the arrhythmogenic phenotype associated with the mutation. This experiment epitomizes the crucial role of non-myocyte cells in cardiac disease.

### 3.3. Modeling the Complex Organization of the Heart in ECTs

Reproducing the anisotropic environment of cells in vitro is essential to obtain a proper function of engineered cardiac tissues. Indeed, in [83], Shum et al. demonstrated that hiPSC-CMs on micropatterned anisotropic sheets diplayed drug-induced arrhythmogenicity, which could not be visualized in regular 2D cultures. Furthermore, a 3D culture environment is needed for cardiomyocytes to develop a phenotype similar to cardiac cells in native tissues, as they are subjected to a constant load, and to develop more physiological cell–cell and cell–ECM contacts [84]. Several different approaches have been developed to reproduce the intricate arrangement of the multiple components of the human myocardium. Those approaches can be divided in two different strategies according to the scaffolding strategy.

#### 3.3.1. Strategies without Scaffold

Designs without scaffold (Figure 1) are based on the observation that cardiomyocytes have an inherent preference to aggregate [85] and can therefore generate matrix-free cardiac tissue constructs. The advantages of such an approach are first that the constructs are independent from potentially immunogenic or pathogenic scaffold materials and that they can reach high cell densitiesm which enhance cell–cell communication. This allows us to recapitulate the features of some diseases where perturbed cellular interactions are central characteristics. The different approaches to building cardiac tissues without a scaffold, their advantages, and their drawbacks are summarized in Figure 1.

The most straight-forward approach consists of building cardiac spheroids. These cellular aggregates can be formed in hanging drops [86,87] to obtain a reproducible size or by self-assembly in nonadhesive wells [73,76,88]. Several protocols have now been set up to obtain multicellular spheroids in a consistent manner without a necrotic core. Those can be formed directly with hiPSCs mixed with other types of cells [89] or with already differentiated cells, which then mature in the spheroids [90,91,92]. We focus on the latter, which represents the most developed technique for now.

Cardiac spheroids are a valuable model of the human heart microenvironment as they can be composed of endothelial cells, cardiac fibroblasts. and cardiomyocytes in proportions similar to those found in a human heart, and they respond to electrical and pharmacological stimulation [87,93]. However, it is difficult to control the patterning of such spheroids and they do not show an anisotropic structure as in human myocardium. One strategy to overcome this limitation is to use bioprinting to assemble cardiac spheroids into complex 3D geometries [73,88,94,95]. Arai et al. built a tubular 3D construct from spheroids composed of hiPSC-CMs, native human dermal fibroblasts (NHDFs), and endothelial cells (ECs) in different ratios by assembling them on a needle array [94]. After one day, they observed a fusion of the spheroids, the construct could be removed from the array after 7 days, and synchronous beating was observed in the tissue. High cell density multicellular microtissues with defined shapes was also recently fabricated by Daly et al. [88] by translating spheroids using a shear-thinning hydrogel with self-healing properties, which enables the user to place and hold the spheroids in the desired shape and can be removed once the cardiac tissue is formed by fusion of the spheroids. This technique allowed Daly et al. to build ring-shaped muticellular cardiac microtissues with high cell densities and to assess their functional characteristics. The pre-assembly of cells in spheroids before forming tissues enables the secretion of ECM, which serves as “glue” and induces better cohesion when the spheroids are assembled in tissues. However, due to the uncontrolled microstructure, the mechanical properties of such tissues are less predictable than with tissues with a defined architecture [96].

Another approach to building 3D scaffold-free constructs is the layer-by-layer technique, which consists in producing organized multicellular cardiac films and assembling them. This can be achieved with temperature-responsive culture surfaces [97] or by filtration [78]. Shimizu et al. formed cell sheets with neonatal rat cardiomyocytes seeded on poly(N-isopropylacrylamide)-grafted surfaces [97]. Those sheets were then detached by reducing the temperature of the medium and overlaid to form a 3D cardiac construct that showed spontaneous beating. This has then been reproduced with the addition of endothelial cells, which enhanced neovascularization of the cardiomyocyte sheets, preserved during cell harvest from the dishes [98]. Moreover, the substrates can incorporate nanotopographical cues to increase cell sheet organization [99]. Finally, the adhesion between cells can be further enhanced by cell surface engineering [100] or nanofilm coating onto single cell surfaces [78]. Nevertheless, those scaffold-free tissues often stay fragile, and a downside of this technique is the difficulty applying a mechanical load on those constructs.

#### 3.3.2. Strategies with Scaffolds

Approaches with scaffolds use an exogenous matrix to mimic the ECM, so as to orient the cardiomyocytes and to recreate the natural 3D environment adequate for cell and tissue growth. Indeed, it has been shown that the ECM regulates the maturation and shape of cardiomyocytes by submitting them to an external load and coordinates myofibril assembly. Hence, the ECM and its fibrillar structure play prominent roles in the control of tissue architecture and contractile strength [101]. The scaffold used to mimic the natural ECM can be of different natures but should fulfill the following requirements: give the cells an environment close to native cardiac ECM; enhance the electromechanical coupling inside the tissue; provide elasticity and mechanical strength to the tissue; and in the case of cardiac repair, should include vasculature [102]. To achieve these requirements, researchers can tune several parameters of the scaffold. General physical characteristics such as porosity, density, or electrical conductivity, which are important aspects of native tissues, must be recreated in engineered scaffolds to guide self-assembly. Mechanical characteristics (stiffness, maximum load, maximum extension, and anisotropy) are also crucial to ensure stable contractility [103]. Furthermore, biochemical properties such as degradation kinetics, toxicity, or the addition of bioactive molecules such as growth factors should be considered and optimized, especially for engineered tissues meant for implantation.

The material as well as its 3D structure should consequently be carefully selected to obtain the desired macro- and micro-environments [104]. To achieve this final goal, different techniques, summarized in Figure 2, were set up.
**Cell-seeding on decellularized extracellular matrix:**As the ECM is a highly complex structural and functional environment that determines cell organization and function, decellularized ECMs have been considered ideal biomaterials in tissue engineering as they offer a solid organized scaffold [105]. This way, the extracellular matrix complex composition, bioactivity, architecture, and vascular tree are kept intact [106] and cells can be reseeded in the scaffold obtained. The decellularization protocols can be enzymatic, chemical, or physical and have been reviewed elsewhere [107]. Decellularized ECMs (dECMs) have been often used as myocardial slices, seeded with hiPSC-CMs and potentially stromal cells to form cardiac patches [108]. In [106], Guyette et al. obtained a more 3D physiological model by reseeding fibers with hiPSC-CMs, which then showed spontaneous contraction. Finally, experiments of whole heart recellularization have been carried out [109].However, the success of this approach in order to build cardiac constructs is largely dependent on the quality of the decellularization, which remains variable across the samples. Moreover, processes tend to be long and a compromise has to be made between the removal of all cells and the preservation of ECM integrity [110]. Additionally, as the ECM surrounds the cells, keeping the intricate ECM network intact may impede proper cell seeding of decellularized material. Therefore, to further maximize cell seeding, ECM postprocessing protocols have been developed and the ECM can now also be solubilized in hydrogels employed as a biological ink in bioprinting [111].**Seeding in liquid hydrogels:**Hydrogels are 3D networks of polymers that can absorb large amounts of fluids, in general, water. Their biocompatibility, good diffusion properties, and high permeability for oxygen, nutrients, and other water-soluble metabolites make them suitable scaffolds for tissue engineering [112]. Moreover, one can control their microscale structure and can tune their mechanical properties to mimic the native ECM by introducing crosslinks and by playing on the polymer molecular weight or density [102]. The main asset of hydrogel scaffolds in tissue engineering is the fact that cells can be mixed into liquid hydrogels, allowing for homogeneous seeding throughout the scaffold, and the mix can be cast to tailor the tissue geometry for the desired application. Once the hydrogel polymerizes, it encapsulates the cells to obtain a tissue with a complex geometry and 3D cell–cell interactions. Hydrogels also enable minimally invasive delivery for cardiac repair: they have already been used for direct injection with hiPSC-CM to regenerate ischemic myocardium and allowed for better cell retention and graft viability [20,107].Several materials are used for the fabrication of hydrogel scaffolds for cardiac tissue engineering. They can be natural polymers such as fibrin, collagen, gelatin, chitosan, hyaluronic acid (HA), matrigel, or even directly dECM, as explained earlier. These natural polymers all present an inherent biocompatibility and biodegradability and can mimic components of the ECM. However, they present batch-to-batch variability and are therefore more difficult to control [113]. Synthetic polymers have also been used to build engineered cardiac tissues; the most commonly used are poly(ethylene glycol) (PEG), poly(vinyl alcohol) (PVA), and poly(2-hydroxyethyl methacrylate) (PHEMA) [114]. Those synthetic polymeric possess more reproducible physical and chemical properties but generally show lower biocompatibility and cell attachment. Cell adhesion can however be improved by molecular modifications and the addition of short peptides to mimic aspects of native cell–ECM interactions [115].To obtain a cardiac-specific anisotropic structure, ECTs made with hydrogel scaffolds can be submitted to physical cues. Mechanical stimulation has been proven to improve the contractility of engineered heart tissues and to result in enhanced maturation of hiPSC-CMs, with better sarcomere organization, improved cell–cell junctions and t-tubules network, as well as higher amplitudes for twitch force and action potentials [116]. Various types of mechanical stimulation have been studied. The most common technique is passive stretch, by seeding hiPSC-CM (potentially mixed with other cells) between two PDMS posts [80,117], but a cyclic stretch, imitating the successive filling and emptying steps of the cardiac cycle, can also further enhance the maturation and structure of the ECTs, as shown in [118]. Several studies also demonstrated that electrical stimulation induces higher force generation and improved alignment and cell–cell junction in ECTs [119,120,121]. Finally, combinations of mechanical and electrical stimulation have been implemented on ECTs, as these signals are coupled in the native heart tissue; these lead to improved functional properties compared to electrical or mechanical stimulation only [122,123]. For more examples, one can refer to other recent reviews [18,116]. Bioactive molecules can also be added in the hydrogel to regulate cell adhesion, proliferation, or differentiation or to improve vascularization [114].**Microfabrication of solid materials:**The scaffold can be patterned to better control the 3D environment of the cells and to make it more reminiscent of the endogenous cardiac tissue. This patterning can also enhance vascularization and improve cell survival when implanted in vivo. To achieve this, the fibrillar aspect of the ECM should be reproduced, and this requires complete control over the micron- to nanometer-scale features of the scaffold [124]. Several technologies of micro- and nano-fabrication have been investigated in that prospect such as solvent casting/particulate leaching, gas foaming, freeze drying, thermally induced phase separation, photolithography, electrospinning, or 3D printing. Those have been comprehensively reviewed elsewhere [124,125]. Those techniques allow us to control the chemistry, topography, elasticity, and conductivity of the scaffold. Among those processes, electrospinning and 3D bioprinting currently experience fast development. Electrospinning is a process that consists in using electrostatic forces to produce nanofibers from a polymer solution [126]. The fibers are deposited on a collector to form a nanoporous scaffold with a tailored shape composed of fibers ranging from the micro- to nano-scale and showing morphological similarities with ECM and high porosity with variable pore-size distribution. Moreover, the fibers can be functionalized with different ECM-derived proteins [127]. These properties are very attractive for tissue engineering. Indeed, in [128], iPSC-CMs were seeded on electrospun dextran vinyl sulfone (DVS) fibers with varying biochemical, architectural, and mechanical properties, and it was shown that the optimal configuration was fibronectin-functionalized DVS matrices with highly aligned fibers and low stiffness. The tissues generated on these scaffolds even demonstrated better maturation than micropatterned fibronectin lines. Many other materials and geometries were tested in other studies, and the electrospun scaffold always improved cell alignment and cardiac function [129]. Moreover, electrospinning is a technology that can be up-scaled for industrial application, which provides hope for potential off-shelf scaffolds for cardiac repair [130].Bioprinting is a layer-by-layer additive manufacturing technology that allows the user to print biological material with a defined pattern [131]. Usually, bioinks consist of previously described hydrogels seeded with cells, in which co-factors can be added. This technology allows for a greater complexity in composition and a higher spatial resolution, as it has enabled to print several inks for the same tissue. Indeed, in [132], a 3D bioprinting method based on extrusion was developed to print collagen using freeform reversible embedding of suspended hydrogels. Components of the human heart at various scales could then be reproduced. A human cardiac ventricle model could even be printed, using collagen ink to form external shells and a high-concentration cell ink composed of human embryonic stem cell-derived cardiomyocytes (hESC-CMs) and 2% cardiac fibroblasts between the walls. This ventricle showed synchronized contractions after 7 days, anisotropic action potential propagation, and wall thickening up to 14% during systole. Hence, this method gives encouraging results for the production of complex cardiac models. Several other techniques have been developed for bioprinting and have been thoroughly reviewed in a recent article [131]. Despite all of their advantages, microfabricated scaffolds have limitations: they are less straightforward to make, as they require additional optimization of the material to the chosen technique; they can contain cytotoxic chemical residues; and when the scaffold is fabricated first and the cells are seeded afterwards, it can lead to scarce and inhomogeneous cell infiltration. On the contrary, in the case of bioprinting for which the cells are mixed into the bioink, it can induce cell loss due to method-specific processes.

**Vascularization:** the complex organization of the heart also comprises the vascular network that delivers oxygen and nutrients to the muscle. Without vascularization, the maximum size of a tissue is limited to 100–150 μm by the maximum diffusion length. Hence, this has motivated strategies to promote the formation of blood vessels precursors in engineered tissues: the addition of fibroblasts and endothelial cells or human umbilical vein endothelial cells (HUVEC) improved the formation of blood vessels precursors [73,74]; decellularized ECM presents an already formed and mature network that can be integrated into the host vasculature when implanted [109]; and the addition of angiogenic factors was also shown to improve vascularization of engineered cardiac constructs [133,134,135]. Larger tissues with well-developed vascular networks could then be used for cardiac repair.

## 4. Different Geometries for Different Readouts and Applications

The 3D constructs with diverse compositions, organized with or without scaffolds can be arranged in different geometries suitable for various readouts and applications. Engineering fully functional organs is a complex goal, and building organoids can already help recapitulate a defined function of the organ [136]. These organoids can be either studied in vitro or used to restore or enhance a defined organ function in vivo. Therefore, different geometries would be suitable for distinct applications. For instance, in vitro drug-screening or disease modeling assays require the reproducible generation of small tissues in order to enable experimental repetitions while consuming a moderate number of cells and quantities of reagents. A tissue designed for cardiac repair would need to be larger and scalable in order to produce strong contractile forces. Moreover, additional properties are needed such as biocompatibility, long-term reliability, and safety as well as pre-vascularization to promote graft integration. Figure 3 exposes the readouts that could be obtained and the preferential applications for the different geometries presented in the following paragraphs.

### 4.1. Cardiac Spheroids

Spheroids are spherical multicellular aggregates that can be formed by several methods: (**1**) by self-assembly of floating cells on nonadhesive plates or in rotation systems, which result in different sizes of spheroids, (**2**) by the hanging drop method to obtain a reproducible size, or (**3**) in microfluidic systems [84]. Their diameter is constrained to hundreds of nanometers by the diffusion barrier, so that the oxygen and nutrients can be transported to their core [137]. Cardiac spheroids show spontaneous contractions, which can be studied to evaluate their viability as well as drug effects. Their main asset is that they can be quickly manufactured with a high throughput and small numbers of potentially costly cells [138].

Moreover, they enable us to easily carry out 3D co-cultures by mixing cardiomyocytes derived from iPSCs with cardiac fibroblasts and endothelial cells to obtain a micro-environment closer to the native heart. Indeed in [91], scaffold-free spheroids were formed with a hanging drop system from hiPSC-CMs and cardiac fibroblasts (CFs) with a ratio of 4:1, similar to healthy tissue. Two- and three-dimensional, on the one hand, and single- and co-culture conditions, on the other hand, were compared. It was shown that the beating activity of co-culture spheroids was improved and that the ultrastructural and electrophysiological features were similar to those of fetal cardiomyocytes. It was demonstrated that the use of a 3D co-culture compared to a 2D single-culture avoids CF activation and myofibroblast transformation and therefore more closely mimics the state of a native heart.

By also adding endothelial cells (ECs), Giacomelli et al. further enhanced the maturation of their cardiac spheroids [76]. They showed that spheroids in tri-culture presented an improved sarcomeric structure with T-tublules as well as enhanced contractility, electrophysiology and mitochondrial respiration. Moreover, RNA-sequencing showed upregulated genes linked to efficient and mature contraction for those tissues compared to tissues generated without ECs or CFs or with other types of fibroblasts. The calcium transients in spheroids with CFs and ECs corresponded to more mature tissues. This possibility of a 3D co-culture is attractive as the cross-talk between the different types of cells can be studied but also non-myocyte contributions to some pathologies can be put in evidence. Indeed, in [76], the authors mixed fibroblasts derived from arrhythmogenic hiPSCs to healthy hiPSC-CMs and hiPSC-ECs and they could recapitulate an arrhythmogenic phenotype.

Cardiac spheroids appear as an adequate model for disease modeling. In [82], tri-culture organoids were built by combining either hiPSC-CMs from healthy donors or patients with HCM with cardiac microvascular endothelial cells (HCMECs) and with primary human cardiac fibroblasts (HCFs) in a single-cell suspension at a physiological cell ratio of 3:5:2. A clear difference was observed between the two groups of spheroids in their spontaneous contractile behavior: while healthy spheroids showed a regular beating pattern, HCM spheroids presented arrhythmic patterns, typical for HCM patients. In this study, it also appeared that the physiological-like state of the organoids is dependent on the culture time, as CM migrate during their culture in spheroids and their distribution is time-dependent. In that prospect, spheroids are interesting as they show long-term viability and thus the analysis is often performed 4 weeks after building the spheroids.

The miniaturized multi-well format of this process is also well adapted for drug testing, as they can recapitulate the response of a native myocardium to classical drugs. In [86], spheroids composed of hiPSC-CMs were formed in hanging drops and subjected to different pharmacological agents. As expected, isoproterenol led to an increase in the spontaneous beating rate, while blebbistatin induced a decrease in contraction amplitude before completely stopping the contraction. Doxorubicin, a cancer drug known for its cardiotoxic effects, was also investigated and induced arrhythmias on spheroids. By adding a tri-culture in [87], Polonchuk et al. employed spheroids to investigate the dose-limiting cardiotoxicity of doxorubicin and had an insight into the mechanisms, also involving cardiac fibroblasts and endothelial cells, which led to its toxicity.

Finally, spheroids can be used as blocks to form larger cardiac tissues. In [88], ring-shaped microtissues with spatially controlled cardiomyocyte-to-fibroblast-cell ratios were bioprinted to replicate the structural features of a scarred tissue following myocardial infarction. To reproduce the scarring process, the cardiomyocyte-to-fibroblast ratio was adjusted between the spheroids at 4:1 for healthy ones and at 1:4 for scarred ones. After showing that the functional features of a scar tissue were reproduced, a pro-regenerative microRNA was used to treat scarred tissues, and an increase in contraction amplitude and cardiomyocyte proliferation could be observed.

### 4.2. Cardiac Patches

Cardiac patches are thin sheets composed of cardiomyocytes and potentially other cells, which can be seeded in a scaffold or interconnected to form a layer [139]. Their main advantage lies in their scalability, as cardiac patches of clinically relevant sizes have been built without loss of functional properties: 4 × 4 cm in [140] or 3.5 × 3.4 cm in [141]. Moreover, the developed active and passive contractile forces can be directly measured on those tissues and they can be submitted to mechanical and electrical stimulation. Additionally, unlike for cardiac spheroids, the degree of structural and functional anisotropy of cardiac patches can be controlled. Cardiac patches were built by casting neonatal rat cardiomyocytes and a fibrin-based hydrogel in a PDMS tissue mold containing a Velcro frame [142]. The resulting tissues contained elliptical pores, which induced cell alignment. It was shown that this improved cell and matrix alignment yielded an increase in anisotropy of action potential propagation, a faster conduction velocity, and larger isometric twitch forces. This technique was used in [143] with hESC-CMs, which exhibited an enhanced maturation in patches, through an increased expression of genes involved in cardiac contractile function, longer sarcomeres, higher conduction velocities (up to 25.1 cm/s), and higher maximum contractile force amplitudes and active stresses of cardiac patches (3.0 ± 1.1 mN and 11.8 ± 4.5 mN/mm${}^{2}$ respectively). These are some of the highest values obtained for cardiac patches seeded with hIPSC-CMs or hESC-CMs.

Nevertheless, the measurement and calculation of the passive and active forces depend on the geometry, the material used, and the number of cells seeded, which makes comparison difficult. Thus, there is a need for standardization in the calculation and expression of the active forces developed in different studies; for example, the force can be expressed per input cardiomyocyte.

Thanks to its scalability, this technique can be used for drug screening on the one hand and cardiac repair on the other hand. In [144], hiPSC-CMs and cardiac fibroblasts were coated with ECM proteins and mixed in different ratios with endothelial cells. With an appropriate hiPSC-CMs:CFs:ECs ratio, a modification of the contractile properties could be seen in the presence of E-4031 and isoproterenol. Hence, cardiac patches can be suitable for drug-screening systems.

On the other hand, in [140], the focus was on the generation of large and functional heart tissues for implantation. Cardiac patches were made from the encapsulation of dissociated hiPSC-CMs in hydrogels patches, cast in square PDMS molds with Nylon frames. The technique could be scaled up to obtain 36 × 36 mm cardiac patches, showing synchronous contractions with a preserved electrical phenotype and yielding forces of 17.5 ± 1.1 mN. A proof of concept of implantation was carried out with smaller patches in window chambers in nude mice, and vascularization of the patch by host vessels was observed. Moreover, the implantation on the rat epicardium showed a preserved patch structure and electrical function in vivo and no increase in the incidence of arrhythmias. However, although the results after 3 weeks in vivo indicated robust survival and vascularization of the graft, the graft still lacked electrical integration with the host rat heart. This lack of graft-host electrical integration was reported in other studies [145] and gives a base for improvements to build efficient tissue engineering therapies for heart repair. In [146], large human cardiac muscle patches were fabricated with trilineage cardiac cells derived from human-induced pluripotent stem cells and implanted in swine with myocardial infarction, and showed improvements in cardiac function. Thus, these results provide grounds for further development of cardiac patches as therapies for ischemic hearts.

Finally, it has been shown that an electronic mesh can be added to cardiac patches without compromising the viability of the cells [147]. This way, cellular activities can be recorded and electrical stimulation for synchronization of cell contraction can be remotely controlled. Electroactive polymers containing physiologically relevant factors can also be chosen in order to control their release with the application of an electrical field. In [147], the stem cell chemoattractant stromal cell-derived factor-I (SDF-I) could be loaded into the scaffold and released under an applied voltage and it promoted cell migration in vitro. This proof-of-concept study shows that other positively charged growth factors or drugs could be stored and released with cardiac patches in vivo, enhancing the integration of the patch and its vascularization or even achieving a therapeutic goal.

### 4.3. Cardiac Strips

Cardiac strips or fibers consist in 3D engineered cardiac tissues cultured in rectangular molds in which they attach on two flexible posts [148] or on a wire [149]. Most of the time, cells are embedded in a hydrogel and the mix is pipetted in the mold, where it polymerizes and compacts to form a beating tissue, bending the posts (or the wire) at each contraction. The main advantages of this geometry are that it provides an anisotropic structure to the tissue, and it allows us to easily measure active stress in a noninvasive way.

The first tissues using this geometry were built with neonatal rat heart cells [148,150], and well-developed, aligned cardiac muscles could be formed. Their active force was measured by video analysis of the motion of the posts. It was shown in these studies that this geometry also allowed us to examine the response of the constructs to drugs, as the tissues exhibited a concentration-dependent chronotropic or inotropic response to classical drugs such as chromanol, quinidine, or doxorubicin.

The development of human-induced pluripotent stem cells induced the construction of human cardiac fibers with cardiomyocytes derived from patients’ hiPSCs. Those constructs already recapitulated key physiological cardiac features [117] but their maturation could be further improved with electrical or mechanical stimulation, easily introduced in this type of setup. Indeed in [45], tissues were subjected to electrical stimulation with increasing frequency in time. It was demonstrated under this conditioning that tissues developed a mature ultrastructure, higher active forces, adult-like gene expression, a positive force–frequency relationship, and better calcium handling. In [151], cyclic stretch was imposed on the tissues and led to an increase in the systolic force and a more mature response to β-adrenergic stimulation.

This geometry applies well for human cardiac disease modeling as well as drug screening. Cardiac dysfunction can be modeled with those tissues by taking patient-specific cells: in [152], the first 3D-ECT model of HCM could be developed by forming tissues with hiPSC-CMs derived from a patient with a BRAF mutation, mixed with stromal cells. Compared to healthy ECTs, the BRAF-ECTs displayed a hypertrophic phenotype with increased tissue size and twitch force, with shorter twitch durations and increased contraction and relaxation rates on day 6. However, those differences in twitch force could no longer be seen on day 11, revealing a temporal aspect of disease modeling with tissue engineering. In [72], polygenic left ventricular hypertrophy could also be modeled for the first time after long-term culture of biowire microtissues with electrical conditioning for 8 months, which resulted in noticeable differences in the contractile function between healthy and affected patients. These results highlight the importance of time and maturation in disease modeling. Additionally, a cardiac pathology can be modeled by disrupting a specific function of the tissues: recently, in [153], arrhythmic 3D microtissues were generated by disrupting cell–cell signaling with the addition of methyl-beta cyclodextrin (MBCD) on tissues composed of healthy hiPSC-CMs and fibroblasts. Cardiac strips have also been demonstrated to be efficient tools for drug screening. Indeed, several studies obtained canonical responses for several pharmacological agents [18,154]. High control of the tissue composition also enables a chamber-specific characterization and drug screening. Zhao et al. achieved the production of either atrial or ventricular tissues and characterized the contractile functions of each tissues [72]. By applying different agents to the tissues and by measuring force and calcium transients, they highlighted chamber-specific responses. With their platform, they were able to create composite cardiac tissues containing an atrial and a ventricular end. This allowed for efficient screening of differential responses to agents with chamber-specific actions.

### 4.4. Cardiac Rings

Cardiac rings are 3D-ECTs cast in circular molds with a central pillar around which they consolidate. They can then be transferred around posts to be subjected to passive stretching for maturation or on force transducers [141] to measure the active and passive developed forces. The advantages of this circular geometry are that it induces a more homogeneous force distribution throughout the tissue [85], it is more comparable to the shape of the heart, and tissues can be easily miniaturized for high-throughput screening. Moreover, this geometry allows us to study the occurrence of re-entrant waves, which are markers for arrhythmia [155,156]. Furthermore, Li et al. showed that re-entrant waves can promote maturation in engineered circular cardiac tissues as they induced better structural organization and improved calcium handling and contractile properties by submitting cardiomyocytes to high frequencies without electrical pacing [157].

Similar to cardiac strips, the preferential applications for cardiac rings are disease modeling and drug screening. Indeed, cardiac rings are attractive tools to model arrhythmic diseases. Goldfracht et al. could build tissues with hiPSC-CM-derived patients suffering from two types of inherited arrhythmogenic syndromes: the long QT syndrome type 2 (LQTS2) and catecholaminergic polymorphic ventricular tachycardia type 2 (CPVT2). They could recapitulate their abnormal phenotypes and exhibit the development of re-entrant arrhythmias in those ECTs with electrical stimulation and arrhythmogenic drugs [158]. Additionally, in [159], chamber-specific circular EHTs could be built and their optical mapping analysis exhibited a normal activation pattern for ventricular tissues while different types of re-entrant waves were observed for atrial tissues. This underlines the potential of those atrial circular tissues to model atrial arrhythmias such as atrial fibrillation. Finally, all of the previously described studies obtained physiological responses to different adrenergic agent or arrhythmogenic drugs with circular tissues, which makes them suitable models for drug screening.

### 4.5. Cardiac Chamber

The previously described models of human heart muscles are simplified and do not provide any measure of cardiac performance as a pump. Hence, engineered cardiac chambers are progressively developed by partially recellularizing human hearts [106] or by using fabricated scaffolds [132,160,161].

Guyette et al. partially re-cellularized a whole human heart scaffold with hiPSC-CMs in a custom human heart bioreactor providing coronary perfusion and left ventricular wall mechanical stimulation [106]. After 14 days of culture, they obtained visible contractions of the resulting myocardium under electrical stimulation.

Li et al. also obtained 3D electro-mechanically coupled, fluid-ejecting miniature human ventricle-like cardiac organoid chambers by embedding hPSC-derived ventricular cardiomyocytes and human dermal fibroblasts in a collagen-based hydrogel and cast the mix in a spherical mold around a balloon [161]. The constructs showed a mature ultrastructure and an upregulation of calcium handling, ion channel, and cardiac-specific proteins compared to other types of 3D constructs. Moreover, those constructs could be perfused and pressure–volume loops could be measured, which gave access to clinically important parameters such as ejection fraction, developed pressure, and stroke work. Electrophysiological properties could also be evaluated. Finally, those cardiac organoid chambers showed expected mechanical and electrophysiological responses when subjected to some known pharmacological agents, which is encouraging for their use for drug screening applications.

In an other study [160], nanofibrous scaffolds made of electrospun polycaprolactone and gelatin were seeded with neonatal rat ventricular myocytes or hiPSC-CMs. By inflicting geometrically controlled injuries in those constructs, the development of spiral waves pinned to the controlled defects was observed. These results indicate the feasibility of studying arrhythmogenic heart diseases on those engineered heart chambers.

The pressure developed with these constructs are much smaller than the values observed in human ventricles as their scale is, for the moment, also limited. For the time being, these engineered cardiac chambers remain useful models to investigate overall physiological cardiac function and multicelllar phenotypes in cardiac disease and in response to pharmacological agents, but such tissues could also be scaled up and used for cardiac transplant in the future.

## 5. Limitations and Perspectives of Cardiac Organoids

Exciting progress has been made towards the formation of simplified models of human myocardium to bridge the gap between animal models and clinical trials in drug-screening and disease modeling. Additionally, these 3D engineered cardiac constructs give encouraging results for their future application in cardiac repair.

Nevertheless, cardiac organoids still show much lower forces than the ones developed by human myocardium [162], as shown in Table 1. This is partly due to their immature state. Indeed, the features of most 3D-ECTs made from cardiomyocytes derived from hiPSCs remain those of fetal or neonatal tissues [18]. Better maturation of the tissues can be achieved, first through maturation of the hiPSC-CMs before tissue formation [163,164] with long-term culture [165], biochemical cues [166,167,168], or biomimetic substrates [169]. Second, as explained earlier, adding electrical and/or mechanical stimulation of the tissues can further enhance their maturity [45,116]. However, the stimulation protocols still need to be optimized and standardized to mimic in vivo maturation of the myocardium during development. Indeed, in native cardiac tissue, from the first heartbeat, the increase in heart rate and active strength follows cell remodeling due to the mechanical stimuli of a heart beating and leads to increased mechanical demands, which engenders more sarcomeric remodeling and cell alignment [170].

There are two main axes for further development in tissue engineering: on the one hand, more effort can be provided towards an improved scalability and a better host integration of engineered cardiac tissues for cardiac repair, and on the other hand, miniaturized models should be further developed and standardized for high throughput drug screening.

For cardiac repair, pre-vascularized tissues have been built and integrated to the host vasculature [73,140], but full and mature vascularization has not yet been achieved in vitro, limiting the size of the tissues for long-term viability. Additionally, a lack of graft–host electrical integration was reported in several studies [140,145]. This obstacle could be overcome by the addition of conduction cells in ECTs. With the development of protocols to obtain iPSC-derived Purkinje cells and to obtain a conduction system being in its early stages [171,172], it provides grounds for further development of ECTs that can couple with the host tissue when implanted.

On the other hand, the development of bioreactor platforms has already enabled the production of engineered heart tissues in an automated, standardized, and repeatable way. Bioreactors allow for strict control of the microenvironment of the cardiomyocytes: Zhao et al. recently built a heart-on-chip system in which small tissue strips could be formed and electrically conditioned, and the tissue contractions could be monitored [173]. They obtained well-formed tissues showing a positive force–frequency relationship and post-rest potentiation. Other bioreactors allowing for dynamic and mechanical stretching of engineered microtissues could also be built to further improve the maturation of the tissues [174]. This precise monitoring of the mechanical load exerted on the tissues enabled Pitoulis et al. to recreate the electromechanical events of in vivo pressure–volume loops with in vitro force–length loops [175]. This way, they could assess the effect of pathophysiological load on tissue remodelling. Finally, these heart-on-chip platforms can be used for high-throughput assessment of drug efficacy or cardiotoxicity [176,177]. These devices will be further developed with the fast evolution of micro- and nanofabrication techniques and represent promising tools for disease modeling and drug screening [124].

## 6. Conclusions

Cardiac tissue engineering has received greater attention during the last years and has provided innovative platforms to study heart failure and cardiomyopathies. The combination of pluripotent stem cells technologies and microfabrication methods has already resulted in the generation of engineered cardiac tissues with different shapes and constituents that can reproduce some cardiac functions.

In the future, engineered cardiac tissues could become even more complex and more closely feature native cardiac tissue. Future developements will include their composition, by producing specific cardiomyocyte populations and by organizing them spatially; their scalability with the production of larger tissues allowed by their full vascularization; and their geometry, by giving them the shape of perfused cardiac chambers to assess their performances as pumps.

## Figures and Tables

**Figure 1 biomedicines-09-00563-f001:**
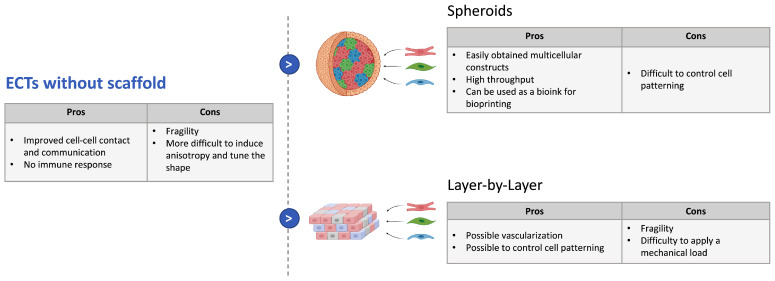
Strategies to build 3D-ECTs without scaffolds, advantages, and drawbacks.

**Figure 2 biomedicines-09-00563-f002:**
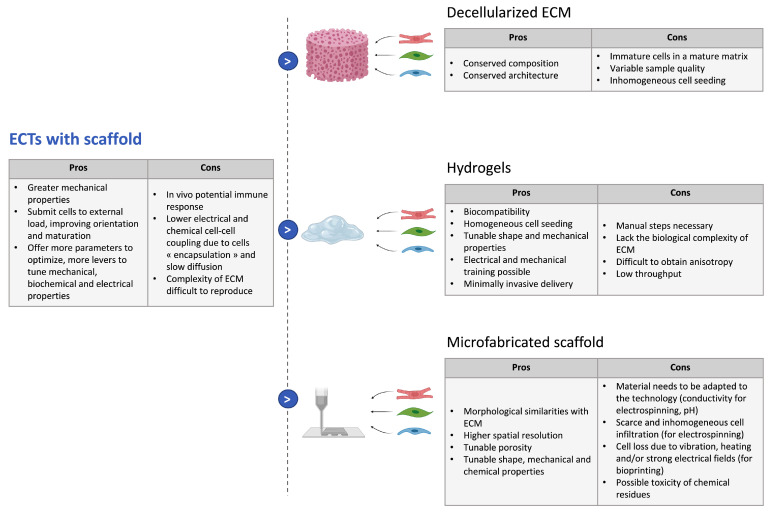
Strategies to build 3D-ECTs with scaffolds, advantages, and drawbacks.

**Figure 3 biomedicines-09-00563-f003:**
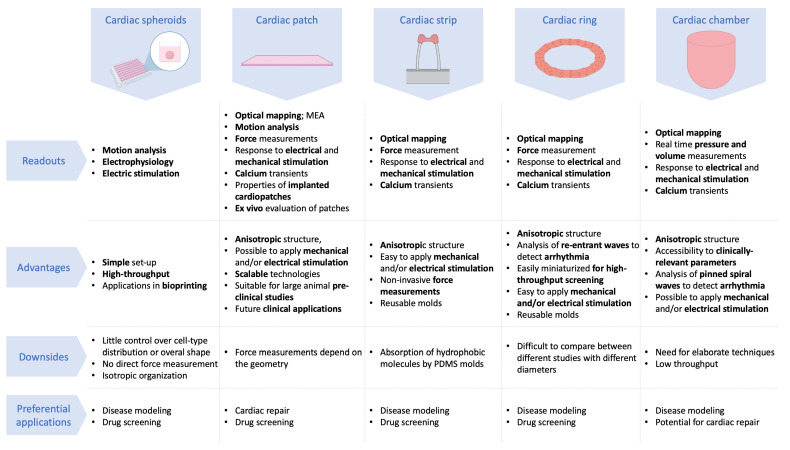
Different geometries for engineered cardiac tissues, their readouts, advantages and downsides, and their preferential applications.

**Table 1 biomedicines-09-00563-t001:** Table summarizing the characteristics of some engineered cardiac tissues from hiPSC-CMs or hESC-CMs in the literature. FFR: Force–Frequency Relationship. ND: Not defined.

Reference	Shape	Cardiac Cells	Supporting Cells	Scaffold	Contractility Performances	FFR	Frank-Starling
Zhang et al., 2013 [143]	Cardiac patch	hESC-CMs	None	Fibrinogen Matrigel	11.8 ± 4.5 mN/mm${}^{2}$	ND	Yes
Shadrin et al., 2017 [140]	Cardiac patch	hiPSC-CMs	None	Fibrinogen Matrigel	13.3 ± 1.0 mN/mm${}^{2}$	Flat to negative	Yes
Turnbull et al., 2014 [117]	Cardiac strip	hiPSC-CMs	None	Bovine collagen type I Matrigel	0.57 mN/mm${}^{2}$	Negative	Yes
Zhao et al., 2019 [72]	Cardiac strip	hiPSC-CMs (ventricular or atrial)	None	Rat tail collagen Matrigel	0.051 ± 0.025 mN/mm${}^{2}$	Positive	ND
Mannhardt et al., 2016 [154]	Cardiac strip	hiPSC-CMs	None	Fibrin Matrigel	0.5 mN/mm${}^{2}$	Flat	Yes
Ronaldson-Bouchard et al., 2018 [45]	Cardiac strip	hiPSC-CMs	Human dermal fibroblasts	Fibrinogen Thrombin	∼2.5 mN/mm${}^{2}$	Positive	ND
Tulloch et al., 2011 [80]	Cardiac strip	hESC-CMs hIPSC-CMs	Stromal	Collagen	0.08 mN/mm${}^{2}$	ND	Yes
Guyette et al., 2016 [106]	Cardiac strip	hiPSC-CMs	None	Decellularized ECM	124.1 ± 94.7 µN	Flat	ND
Goldfracht et al., 2020 [159]	Cardiac ring	hESC-CMs (ventricular or atrial)	None	Bovine collagen	0.92 ± 0.09 mN/mm${}^{2}$ (ventricular) 0.19 ± 0.04 mN/mm${}^{2}$ (atrial)	ND	Yes
Li et al., 2020 [157]	Cardiac ring	hiPSC-CMs	None	None	0.23 ± 0.12 mN/mm${}^{2}$ (0 ReW) 0.54 ± 0.15 mN/mm${}^{2}$ (2 ReW)	ND	Yes
Tiburcy et al., 2017 [141]	Cardiac ring	hESC-CMs	Fibroblasts	Rat tail collagen Matrigel	6.2 ± 0.8 mN/mm${}^{2}$	Positive	Yes
Li et al., 2019 [161]	Cardiac chamber	hESC-CMs	Fibroblasts	Bovine collagen type I Matrigel	Pressure ∼50 μm Hg	ND	ND
Guyette et al., 2016 [106]	Cardiac chamber	hiPSC-CMs	None	Decellularized ECM	Pressure: 2.4 ± 0.1 mm Hg	ND	ND

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
