# Peer review of "Cardiac Organoids to Model and Heal Heart Failure and Cardiomyopathies"

_biomedicines, 2021, doi:10.3390/biomedicines9050563_

Round 1

Reviewer 1 Report

The review is interesting, however therea a few points should be addressed for publication:

-There are also other papers in the literature that are interesting that could be included:

Alternative strategies in cardiac preclinical research and new clinical trial formats.  DOI: 10.1093/cvr/cvab075.

Principles of Spheroid Preparation for Creation of 3D Cardiac Tissue Using Biomaterial-Free Bioprinting. doi: 10.1007/978-1-0716-0520-2_12.PMID: 32207113

Despite mention that several technologies of micro- and nano-fabrication have been investigated and have referred to other recent reviews [122,123], current biofabrication techniques are of great interest and should be included more extensively, for instance, bioreactors and heart-on-a-chip approaches.

Please, explain briefly at the test the disadvantages of the 3D layer-by-layer strategy shown in figure 1.

-As the paper is about organoids, are cardiac patches considered 3D engineered cardiac tissues?

Author Response

We thank the reviewer for the suggestions and thoughtful comments.

-There are also other papers in the literature that are interesting that could be included: 

Alternative strategies in cardiac preclinical research and new clinical trial formats.  DOI: 10.1093/cvr/cvab075.

This was added in reference 15, cited on line 30 in the introduction, on the limitations of animal models, and in line 76, to give a review of the existing models to further improve our understanding of mechanisms leading to cardiomyopathies. 

Principles of Spheroid Preparation for Creation of 3D Cardiac Tissue Using Biomaterial-Free Bioprinting. doi: 10.1007/978-1-0716-0520-2_12.PMID: 32207113

This was added in reference 95, cited on line 238 in the use of bioprinting to assemble spheroids.

Despite mention that several technologies of micro- and nano-fabrication have been investigated and have referred to other recent reviews [122,123], current biofabrication techniques are of great interest and should be included more extensively, for instance, bioreactors and heart-on-a-chip approaches.

This is indeed a very interesting topic. An additional paragraph has been added in the section 4. Limitations and perspectives starting from line 682.

Please, explain briefly in the text the disadvantages of the 3D layer-by-layer strategy shown in figure 1.

There was a mistake in this figure, the table associated to Layer-by-Layer technique is now corrected now corresponds to what is written in the text lines 262-267. The disadvantages are the fragility and the difficulty to apply a mechanical load on those tissues

-As the paper is about organoids, are cardiac patches considered 3D engineered cardiac tissues?
We agree with the reviewer that cardiac patches are not strictly speaking organoids but there are no strict definitions of what could be considered as organoids or not. In this review, we opted for a broad definition in order to provide the readers a complete overview of the different 3D structures that have been developed so far. We believe that cardiac patches represent an important step in the development of cardiac tissues and offer some specificities that are worth reporting to the readership.

Reviewer 2 Report

language should be revised, especially the stream of sentences.

images in figures 1 and 2 are blurry.

more reliable schematic diagram should be given to explain the research work.

is it possible to apply additional analytical methods to prove your results.

Author Response

We thank the reviewer for the suggestions and thoughtful comments.

language should be revised, especially the stream of sentences.

The paper has been edited by an English-native reviewer.

images in figures 1 and 2 are blurry.

The quality of the pictures was upgraded to a 600 dpi resolution.

more reliable schematic diagram should be given to explain the research work.

This work is a state-of-the-art review on cardiac organoids and it is therefore difficult to provide diagrams to explain our approach. We instead provide 3 figures that are reporting the major components leading to the development of 3D cardiac tissues, which could be viewed as schematic diagrams.

is it possible to apply additional analytical methods to prove your results.

This is a state-of-the-art review where we develop on current techniques to develop cardiac tissues. There are as such no specific results generated by this work, but rather a report on the results obtained by the different referenced authors. It will be hard to further develop on methods used to support some of the proposed results, as it will very significantly increase the length of this review. Alternatively, we provide a figure with the major analytical methods that have been used and checked that all works are referenced.